# HierVision: Standardized and Reproducible Hierarchical Sources for Vision Datasets

Tejaswi Kasarla[1], Ruthu Hulikal Rooparaghunath[1], Stefano D'Arrigo[2], Gowreesh Mago[1], Abhishek Jha[3], Melika Ayoughi[1], Swasti Shreya Mishra[1], Ana Manzano Rodríguez[1], Teng Long[1], Mina Ghadimi Atigh[1], Max van Spengler[1], Pascal Mettes[1]

[1] University of Amsterdam, The Netherlands, [2]Sapenzia University of Rome, Italy, [3] KU Leuven, Belgium

## Abstract

*One-vs-rest training is a pervasive optimization regime in deep learning, whether the problem is supervised, self-supervised, or multi-modal in nature. The real world is however not binary, but governed by hierarchies. Hierarchies provide key information about the semantic relation between concepts, about which mistakes to avoid, and about the inherent organization of vision and language itself. Hierarchical learning, therefore, has a long history in computer vision and has gained further traction with the rise of hyperbolic deep learning. Currently, however, hierarchies are not standardized and centrally organized. Instead, such knowledge is scattered around various repositories, with inconsistent formatting, organizations, and availability. The lack of a central hub for hierarchies in vision datasets harms the utility and reproducibility of hierarchical learning. This paper introduces* HierVision*, a central hub for hierarchical knowledge in vision datasets. This hub contains 60+ hierarchical sources, spanning actions, concepts, fine-grained categories, vision-language, and more. We outline a uniform coding of the hierarchies and procedures to embed them in existing pipelines. With this hub, we hope to positively impact the broad use and re-use of hierarchies for deep learning in computer vision. The* HierVision *hub is available at:* `https://github.com/tkasarla/HierVision`

## 1. Introduction

Hierarchies are ubiquitous data structures across all sciences; from lesion taxonomies in the medical domain to animal ontologies in biology and semantic trees in natural language [2, 82]. Such tree-like structures have been used for centuries to organize our data and natural phenomena [1]. Computer vision deals with categorizing concepts from the real world, and datasets are therefore commonly organized hierarchically. Consider for example the WordNet hierarchy behind ImageNet [27], the biological ontology of birds in CUB [117], or the tree of verbs in Kinetics [58].

Despite the widespread availability of hierarchical information for computer vision, such knowledge is typically ignored when training deep networks. Instead, one-versus-rest optimization through cross-entropy and contrastive objectives are default options [22, 50]. Such a setup presents a binary view to categorization, where classes are either positive or (equally) negative. As a result the standard deep learning setup misses crucial hierarchical information about class similarities. The lack of hierarchical usage negatively impacts learning [53, 93, 126], generalization [93], error severity [8], and more.

An important reason for the lack of hierarchical integration in modern deep learning for vision is a geometric mismatch. Deep learning is Euclidean by default. Hierarchies are, however, exponentially growing structures, which leads to distortion when embedding them in Euclidean space [103], as Euclidean volumes grow only polynomially with their radius [88]. Recently, hyperbolic learning has rapidly gained traction in computer vision [80], as a natural space for embedding hierarchies [36, 88, 101, 115] and therefore a natural solution for hierarchical computer vision [3, 5, 29, 57, 74]. As a result, there is a growing demand for hierarchical knowledge in vision datasets.

An important issue currently is that there is no central hub for storing and sharing hierarchies. This does not align with the best scientific practices and hampers research. Not only are hierarchies arbitrarily hard to find depending on the dataset, but they are also not standardized and can even be altered. As such, it is unnecessarily hard to use hierarchies, and reproducibility is low since it is unknown whether hierarchies are identical. This paper introduces *HierVision*, a central hub for sharing hierarchies in vision datasets. Our goal is simple: create a continuous effort to store hierarchies for all vision datasets in a single place. Each hierar-

chy is standardized in a single format for ease of use and reproducibility. The hub also contains pipelines for visualization, analysis, and integration in deep learning and hyperbolic embedding pipelines. With *HierVision*, we want to make the community aware of the broad potential of hierarchical knowledge and the need for a central hub to organize computer vision hierarchically.

## 2. Related Work

### 2.1. Hierarchical Datasets

For clarity, we categorize prominent dataset hierarchies into two groups–those emphasizing semantic ontologies and those following biological taxonomies.

Semantic hierarchies are typically derived from human-defined knowledge bases or lexical resources. A foundational example is the use of WordNet [82], a large lexical database of English, to structure the ImageNet dataset [27]. This provided a rich, structured ontology that has been instrumental in the development of deep learning models. Other prominent datasets include CIFAR-100 [61], PASCAL-VOC [32], and OpenImages [62]. The BREEDS benchmark, derived from ImageNet, explicitly uses the class hierarchy to study robustness [102]. Such semantic structures are not limited to object recognition and extend to domains like medical imaging with datasets like CheX-pert [109] and scene understanding with ADE20K [133].

The second category of datasets follows formal biological taxonomies, providing a scientifically grounded structure for fine-grained visual categorization. These datasets are critical for applications in biodiversity and conservation. For example, iNaturalist [114] organizes species observations according to the taxonomic rank (kingdom, phylum, class, etc.), ensuring that classes have a hierarchical relationship. TreeOfLife-10M and Rare Species [110], Classic fine-grained benchmarks CUB [117] and NABirds [113] are built on the taxonomy of species and genera, and datasets like AutoArborist [7] structure tree images by botanical taxonomy.

### 2.2. Hierarchies enhance vision tasks

The use of hierarchies as a source of prior knowledge is a long-standing concept in computer vision [27, 66, 78]. Classical approaches from the pre-deep learning era explicitly modeled the compositional nature of objects and scenes. Part-based approaches, such as pictorial structures [35] and grammar-based models [135], organize objects and scenes into parts to improve image recognition and interpretability. [34, 35, 75, 107, 135].

In the modern deep learning era, hierarchical knowledge is integrated through various mechanisms such as class taxonomies, structured loss functions, and specialized architectures [8, 9, 38, 85]. These approaches improve many tasks such including image classification [20, 53, 93], action classification [43, 74], and robustness to distribution shifts [102]. Hierarchical approaches were also used to measure the severity of classification mistakes, where misclassifying an object as a close relative in the hierarchy is penalized less severely [8, 9, 38, 39]. Furthermore, hierarchical information has been successfully incorporated into contrastive learning [13, 14, 46], and vision-language models [37, 90]. The utility of hierarchical methods also extends to applied domains such as medical imaging [19] and autonomous driving [83]. A common theme of these works is their reliance on Euclidean geometry to model these hierarchical relationships.

### 2.3. Hyperbolic learning

Hyperbolic learning has emerged as a powerful paradigm for encoding and exploiting hierarchical relationships in visual data. Owing to the constant negative curvature of hyperbolic space, it can be thought of as a continuous version of a tree, making it a good choice to accommodates tree-like structures while preserving distances [48, 112]. In recent years, hierarchical embeddings have been performed in hyperbolic space [88], leading to successfully embedding complex trees with low distortions [36, 63, 89, 101, 115, 129].

Many computer vision tasks inherently involve hierarchies, for example, semantic grouping or biological taxonomies (Sec. 2.1). A wide range of works have recently shown the potential and effectiveness of using a hyperbolic embedding space for both supervised and unsupervised learning [80]. Specifically, in supervised settings, the hierarchical prior knowledge of the datasets can be embedded in hyperbolic space, after which the visual representations can be mapped to the same space and optimized to match this hierarchical organization [5, 57, 74]. Hyperbolic embeddings have shown benefits in classification [41, 45, 116], segmentation [3, 18], out-of-distribution detection [36, 57, 116], uncertainty quantification [3, 18], zero-shot learning [4, 51, 69], continual learning [5, 25, 111], hierarchical representation learning [29, 30, 72, 74], contrastive learning [40, 130], generative models [64, 95], and vision-language models [28, 54, 92, 94]. Recently Ayoughi *et al*. [6] discussed optimal tree structure for hyperbolic embeddings.

## 3. Hierarchies

While the benefits of hierarchical information in computer vision are well-established, its practical adoption has been limited by fragmented and inconsistent hierarchy management across datasets. Hierarchies exist in various formats, from simple text files and custom XML/JSON to folder-based organizations, which require custom parsing for each data set. This fragmentation hinders reproducibility and the

development of hierarchical computer vision methods. To address this, we introduce *HierVision*, a centralized hub that standardizes hierarchy representation and enables seamless integration with existing tools. In the following sections, we first define a common hierarchy format that can be used with graph processing library like NetworkX [47] (Section 3.1). Then we briefly discuss our standardization process (Section 3.2. We then discuss in detail the current datasets (Section 3.3) and finally present some dataset hierarchy statistics (Section 3.4).

### 3.1. Hierarchy format

To enable standardized storage and use of hierarchical information across diverse vision datasets, we adopt a uniform graph-based representation format. Each hierarchy is modeled as a directed, rooted tree encoded in JSON. Formally, we represent a hierarchy as a graph, $G = (V, E)$, where $V$ is the set of nodes (*e.g.* dataset classes or parents) and $E \in V \times V$ is the set of directed edges, such that an edge $(u, v) \in E$ denotes a parent-child relationship and node $v$ is a subclass of node $u$. In practice, each hierarchy is stored in a JSON with the following components:

- `"nodes"`: A list of nodes, each with an integer `"id"` and a human-readable `"label"`. These define the concepts in the hierarchy.
- `"links"`: A list of directed edges, each represented as a with `"source"` and `"target"` keys indicating parent and child node IDs, respectively.
- `"directed"`: A boolean flag, always set to `true`.
- `"multigraph"`: A boolean flag, always set to `false`, enforcing at most one edge between any pair of nodes.

An excerpt of a simplified hierarchy is shown below:

```
{
    "directed": true,
    "multigraph": false,
    "nodes": [
        {"id": 3, "label": "root"},
        {"id": 2, "label": "animal"},
        {"id": 1, "label": "dog"},
        {"id": 0, "label": "cat"}
    ],
    "links": [
        {"source": 3, "target": 2},
        {"source": 2, "target": 0},
        {"source": 2, "target": 1}
    ]
}
```

Listing 1. A sample JSON format of simple hierarchy. All hierarchies in *Hiervision* follow this structure, allowing easy visualization and use in downstream applications

In this example, `"root"` is the global ancestor of all nodes, `"animal"` is a direct child of `"root"`, and both `"cat"` and `"dog"` are subclasses of `"animal"`. This structure is fully compatible with standard graph-processing libraries, particularly NetworkX [47], which we

use throughout our implementation. Hierarchies can be directly loaded as `networkx.DiGraph` objects, enabling efficient hierarchy traversal, visualization, validation, and integration into hierarchical deep learning pipelines, including those that rely on hyperbolic embeddings or hierarchy-aware loss functions.

### 3.2. Standardizing the dataset hierarchies

We collect hierarchies from over 61 vision datasets spanning various domains such as object recognition, fine-grained classification, action understanding, scene interpretation, video analysis, and medical imaging. These hierarchies originate from either the dataset creators themselves or other papers that construct or refine hierarchies of existing datasets for specific tasks. Across these sources, we observe significant variation in format: some hierarchies are in graph-structured files (e.g., JSON, XML trees). Others use flat lists with indentation or prefix-based identifiers to imply structure. Some are only documented visually or embedded in figures or tables in papers. These sources vary significantly in format, structure, and accessibility, requiring a standardization process.

To unify these into a standardized format, we add each hierarchy into the JSON graph structure described in Sec. 3.1. We parse and extract the node and edge structure. We resolve any label inconsistences and add a single root node. We validate the resulting graph using NetworkX to ensure it forms a connected, directed tree.

In cases where multiple versions of a hierarchy exist (e.g., fine vs. coarse levels), we store each version explicitly. This ensures that downstream tasks can choose the level of abstraction best suited to their needs.

### 3.3. Dataset coverage

Our *HierVision* hub currently covers over 50 vision datasets spanning a wide range of domains and recognition tasks. The collection includes image classification datasets such as CIFAR-100, ImageNet-100, ImageNet-1K, ImageNet-21K, and iNaturalist; semantic segmentation datasets including ADE20K and Cityscapes; action recognition datasets such as Kinetics, ActivityNet, and FineSports; fine-grained categorization like CUB, FGVC-Aircraft, and TreeOfLife-10M; and medical imaging datasets like CheXpert and DeepLesion. Hierarchies in these datasets range from shallow groupings of a few categories to deeply nested structures with thousands of classes, reflecting both semantic and biological taxonomies. Further details, including taxonomy type and hierarchy source, are summarized in Table 1.

Figure 1 shows visualizations of the hierarchical structures for six representative datasets: CIFAR-100 (semantic object classification with a two-level hierarchy), Cityscapes (urban scene segmentation with a class grouping tree), RareSpecies (a deeply nested biological taxonomy), Ac-

tivityNet (action recognition with a hierarchical verb ontology), Moments in Time (an action dataset with a flat class structure), and COCO-Stuff-10k (scene parsing with a multi-level segmentation hierarchy). These examples highlight the diversity in both the structure and scale of the hierarchies, ranging from compact, balanced trees to large, irregular graphs with hundreds or thousands of nodes.

**ImageNet sources.** For ImageNet-100, ImageNet-1K,, we reproduce the hierarchies as published by their sources [6, 27, 67]. We do not alter labels, add or remove edges, or resolve multiple inheritance beyond what is fixed upstream; we re-encode the published structure in our JSON.

### 3.4. Hierarchy Statistics

We visualize statistics across all collected hierarchies to assess their structural diversity. Figure 2 summarizes four key aspects: node count, maximum depth, the relationship between node count and depth, and average branching factor.

Most hierarchies contain fewer than 1,000 classes, though there is a long tail of large-scale taxonomies such as TreeOfLife-10M [110], ImageNet-21K [98], and Bamboo [132], that reach tens or hundreds of thousands of nodes (Figure 2a).

The majority of hierarchies are shallow, with depths of 2 or 3, as seen in datasets like CIFAR-100 and ADE20K (Figure 2b). However, a subset of datasets such as TreeOfLife-10M [110] and Visual Genome [60] feature deeply nested structures, with depths exceeding 10. The scatter plot of node count versus maximum depth (Figure 2c) highlights this diversity: some datasets combine high node counts and depth, while others are both small and shallow.

Average branching factor calculates the average number of children for each node, which also varies widely (Figure 2d). Biological taxonomies tend to be deep and balanced, with low branching factors (1–3), while semantic hierarchies such as ADE20K [133] and Objects-365 [104] are broad and flat, with extremely high branching at the root.

### 3.5. Licensing & Maintenance

We redistribute *hierarchical metadata only* (class names and edges), never images or videos. For each dataset we cite the license and source (URL/DOI). To support reproducibility, we maintain semantic versioning at the hub level, at the per-dataset hierarchy level, and for the JSON schema used to encode them. Releases include a changelog, source citations, and integrity checks. Hierarchies are distributed in a JSON graph format; users can compress or convert very large files to Parquet, HDF5 locally if desired.

## 4. Integration into Deep Learning Pipelines

The standardized graph-based format adopted by *HierVision* enables seamless integration of hierarchical information into modern deep learning workflows. In this section,

we outline typical approaches for incorporating hierarchies, ranging from data loading and label preprocessing to advanced hierarchical loss design and representation learning.

Hierarchies stored in our JSON format can be loaded directly using widely adopted graph libraries such as NetworkX:

```
import json, networkx as nx
with open('hierarchy.json') as f:
    graph_dict = json.load(f)
G = nx.node_link_graph(graph_dict)
```
Listing 2. Loading a hierarchy into `networkx`.

Once loaded as a graph object, the hierarchy can be queried to obtain ancestor or descendant sets for each class, compute semantic distances between nodes, or extract subhierarchies for specialized tasks. Below, we discuss the relevance of hierarchies to deep learning in euclidean space (Section 4.1) and hyperbolic space (Section 4.2. Additionally in hyperbolic learning, we show how the few of the hierarchies can be embedded into hyperbolic space.

### 4.1. Relevance to Hierarchy-Aware Supervision

#### 4.1.1. Hierarchy-Aware Label Representations and Multi-Level Outputs

*HierVision* enables augmenting dataset labels with hierarchical context by allowing straightforward retrieval of parent and ancestor labels for any fine-grained class. This facilitates multi-task learning, where models predict class probabilities at multiple hierarchical levels (e.g., object category and super-category) simultaneously, typically via auxiliary output heads and backpropagating loss at each level [118, 126]. Such coarse-to-fine supervision guides feature learning: high-level layers discern broad distinctions, while deeper layers refine for fine-grained classification.

Alternatively, hierarchical classification [106] predicts sequentially, predicting a coarse category before specializing to specific subclasses [17, 70, 93, 118]. In both approaches, *HierVision*'s standardized graph simplifies retrieving relevant ancestor or child classes, ensuring hierarchy-consistent predictions and providing interpretable outputs at multiple levels of detail.

#### 4.1.2. Hierarchy-Aware Loss Functions

*HierVision* facilitates the design of loss functions that account for inter-class relationships, moving beyond standard cross-entropy's uniform error penalty. By leveraging the hierarchy, errors can be weighted by their distance in the tree or reflected in structured objectives.

A classic example is hierarchical softmax [85], which factors the prediction over a tree of classes. Instead of a

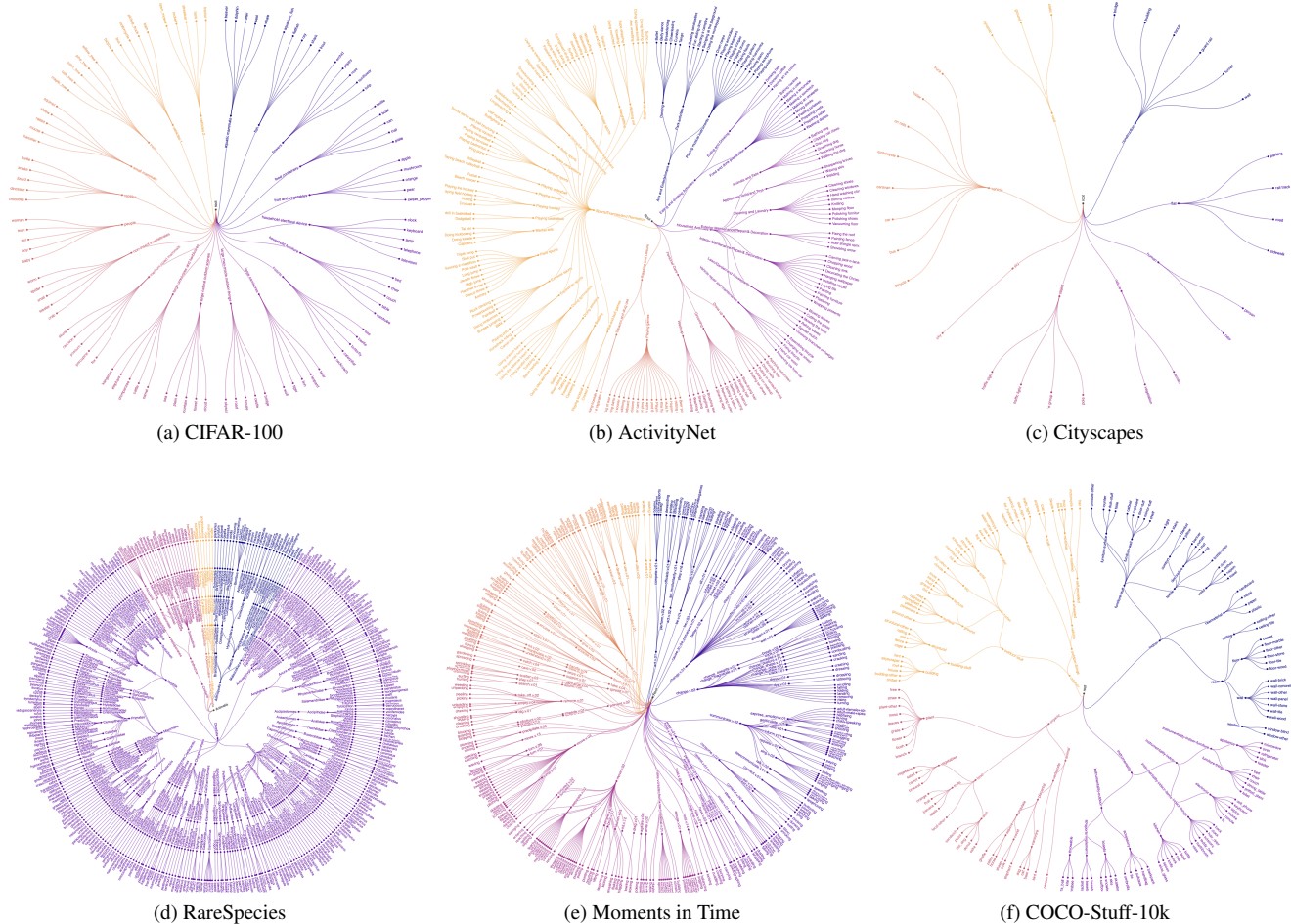

Figure 1. Hierarchy visualizations for a selection of datasets form *HierVision*. The coverage in the repository hub ranges from small number of classes and shallow trees to very large number of classes and deep trees.

flat $N$-class prediction, the model predicts a path from root to leaf, decomposing the problem into a sequence of smaller binary classification tasks. For a tree with internal nodes $\mathcal{N}$, let $y$ denote the true class and $\mathbf{z}$ the logits. The hierarchical softmax decomposes the probability of class $y$ as:

$$P(y \mid \mathbf{z}) = \prod_{n \in \mathrm{path}(y)} P(n \mid \mathrm{parent}(n), \mathbf{z}) \qquad (1)$$

where $\mathrm{path}(y)$ is the sequence of nodes from the root to $y$. The loss is then computed as the negative log-probability along this path:

$$\mathcal{L}_{\mathrm{hier-softmax}} = -\log P(y \mid \mathbf{z}) \qquad (2)$$

This approach not only reduces computational cost for large $N$ but also implicitly aligns internal representations with the taxonomy.

Beyond hierarchical softmax, cost-sensitive losses penalize mistakes according to taxonomy distance, e.g., based on the length of the shortest path or the depth of the lowest common ancestor (LCA) between the predicted class $\hat{y}$ and true class $y$ [8]. Let $d(y, \hat{y})$ denote the tree distance between classes. The loss can be defined as:

$$\mathcal{L}_{\mathrm{hier-LCA}} = \sum_{i=1}^{N} d(y_i, \hat{y}_i) \cdot \ell(y_i, \hat{y}_i) \qquad (3)$$

where $\ell$ is the standard loss (e.g., cross-entropy), and $d(y, \hat{y})$ is typically the path length or a normalized form thereof.

Similarly, hierarchy-based label smoothing replaces one-hot targets with soft distributions, assigning higher probability mass to classes near the ground truth in the hierarchy [8, 97, 99]. For each target $y$, define the smoothed label $\tilde{y}$ as:

$$\tilde{y}_j = \frac{\exp(-\lambda \cdot d(y, j))}{\sum_{k=1}^{N} \exp(-\lambda \cdot d(y, k))} \qquad (4)$$

where $d(y, j)$ is the distance in the hierarchy between $y$ and $j$, and $\lambda > 0$ is a hyperparameter controlling the smooth-

Table 1. All hierarchies currently available in *HierVision*. Datasets with multiple hierarchy versions (e.g., coarse/fine) are marked with * *(reported version)*. If the hierarchy was sourced from another paper, it is cited in the "Hierarchy Source" column.

| | Dataset | Hierarchy Source | Original Format | Nodes | Edges | Depth | Classes |
|---|---|---|---|---|---|---|---|
| **Actions / Video** | ActivityNet [15] | - | JSON | 245 | 244 | 3 | 200 |
| | CMU MoCap [26] | - | WEB | 306 | 305 | 3 | 280 |
| | FineSports [124] | - | PKL | 65 | 64 | 2 | 52 |
| | HDM05 [86] | - | JSON | 26 | 25 | 2 | 20 |
| | HowTo100M [81] | - | JSON | 142 | 141 | 2 | 129 |
| | HumanAct12 [44] | - | NPY | 47 | 46 | 2 | 34 |
| | Mini-Kinetics-200 [58, 123] | - | JSON | 240 | 239 | 3 | 200 |
| | MIntRec* *(all categories)* [131] | - | TSV | 25 | 24 | 2 | 22 |
| | Moments in Time [84] | - | JSON | 486 | 485 | 4 | 339 |
| | Pseudo-Adverbs (ActivityNet) [31] | - | CSV | 758 | 757 | 2 | 643 |
| | Pseudo-Adverbs (MSRVTT) [31] | - | CSV | 571 | 570 | 2 | 464 |
| | Pseudo-Adverbs (VATEX) [31] | - | CSV | 1686 | 1685 | 2 | 1550 |
| | Something-Something V2* *(coarse to fine)* [42, 76] | [76] | JSON | 225 | 224 | 2 | 174 |
| | UCF101 [108] | - | CSV | 127 | 126 | 3 | 101 |
| **Biological** | AwA2 [120] | - | TXT | 86 | 85 | 3 | 50 |
| | BioTrove-Balanced [127] | - | CSV | 826 | 825 | 7 | 300 |
| | BioTrove-LifeStages [127] | - | CSV | 19 | 18 | 6 | 5 |
| | BioTrove-Unseen [127] | - | CSV | 2497 | 2496 | 6 | 1672 |
| | CUB-200-2011 [117] | - | JSON | 251 | 250 | 3 | 200 |
| | iNaturalist [114] | - | CSV | 4214 | 4213 | 2 | 4200 |
| | MammalNet [21] | - | TSV | 260 | 259 | 3 | 173 |
| | Marine Tree [11, 12] | - | CSV | 79 | 78 | 5 | 62 |
| | NABirds [113] | - | TXT | 1011 | 1010 | 4 | 555 |
| | Rare Species [110] | - | CSV | 1024 | 1023 | 6 | 400 |
| | Tree of Life [110] | - | CSV | 635463 | 635462 | 22 | 537235 |
| | VegFru-Fru92 [52] | - | JSON | 103 | 102 | 2 | 92 |
| | VegFru-Veg200 [52] | - | JSON | 216 | 215 | 2 | 200 |
| **Medical** | CheXpert [55] | - | CSV | 15 | 14 | 3 | 11 |
| | DeepLesion [125] | - | CSV | 172 | 171 | 5 | 117 |
| | MIMIC-CXR [55] | - | CSV | 15 | 14 | 3 | 11 |
| | OpenCell [23] | - | CSV | 17 | 16 | 3 | 11 |
| **Objects / General** | Bamboo [132] | - | JSON | 298307 | 298306 | 10 | 285340 |
| | Caltech-101 [65] | - | Folder | 112 | 111 | 4 | 102 |
| | CIFAR-100 [61] | - | PKL | 121 | 120 | 2 | 100 |
| | COCO-10K [16] | [3] | JSON | 234 | 233 | 8 | 171 |
| | COD10K [33] | - | JSON | 75 | 74 | 2 | 69 |
| | CORe50-Balanced [71] | [5] | TXT | 70 | 69 | 4 | 50 |
| | CORe50-Unbalanced [71] | [5] | TXT | 66 | 65 | 5 | 50 |
| | EgoObjects [134] | [5] | JSON | 1665 | 1664 | 14 | 1179 |
| | FGVC-Aircraft [77] | - | TXT | 201 | 200 | 3 | 100 |
| | Fashionpedia-Attributes [56] | - | JSON | 306 | 305 | 2 | 294 |
| | Fashionpedia-Categories [56] | - | JSON | 62 | 61 | 3 | 46 |
| | IP102 [119] | - | TXT | 113 | 112 | 3 | 102 |
| | ImageNet-100* *(ImageNet-100)* [27] | [67] | Folder | 121 | 120 | 2 | 100 |
| | ImageNet-1K [27] | - | Folder | 1778 | 1777 | 8 | 1343 |
| | ImageNet-21K* *(ImageNet-21K-P)* [98] | - | Folder | 74402 | 74401 | 19 | 57919 |
| | ImageNet-OOD* *(ImageNet-21K-P)* [128] | [98] | Folder | 844 | 843 | 3 | 634 |
| | MAdVerse [100] | - | JSON | 656 | 655 | 4 | 578 |
| | Matador [10] | - | WEB | 82 | 81 | 5 | 59 |
| | Objects365 [104] | - | WEB | 377 | 376 | 2 | 365 |
| | OpenLORIS [105] | [5] | JSON | 98 | 97 | 7 | 69 |
| | OpenImages [62] | - | JSON | 602 | 601 | 5 | 525 |
| | PASCAL VOC [32] | [3] | XML | 36 | 35 | 6 | 21 |
| | Portrait Mode 400 [49] | - | CSV | 446 | 445 | 3 | 400 |
| | Stanford Cars [59] | - | Folder | 206 | 205 | 2 | 196 |
| | Stanford Online Products [91] | - | Folder | 22634 | 22633 | 2 | 22622 |
| **Scenes / Places** | ADE20K* *(scene graph)* [133] | [3] | JSON | 1116 | 1115 | 2 | 1105 |
| | Cityscapes [24] | - | WEB | 39 | 38 | 2 | 30 |
| | DLD3V-10K [68] | - | JSON | 81 | 80 | 2 | 64 |
| | Grocery [79] | - | CSV | 125 | 124 | 2 | 81 |
| | Mapillary Vistas [87] | - | JSON | 81 | 80 | 3 | 66 |
| | Million-AID [73] | - | XML | 74 | 73 | 3 | 51 |
| | PACO-EGO4D [96] | - | JSON | 515 | 514 | 2 | 441 |
| | PACO-LVIS [96] | - | JSON | 532 | 531 | 2 | 458 |
| | SUN360 [122] | - | WEB | 378 | 377 | 3 | 356 |
| | SUN397 [121] | - | CSV | 418 | 417 | 3 | 397 |
| | SUN908 [121] | - | WEB | 925 | 924 | 3 | 904 |
| | Visual Genome [60] | - | JSON | 10503 | 10502 | 18 | 6114 |

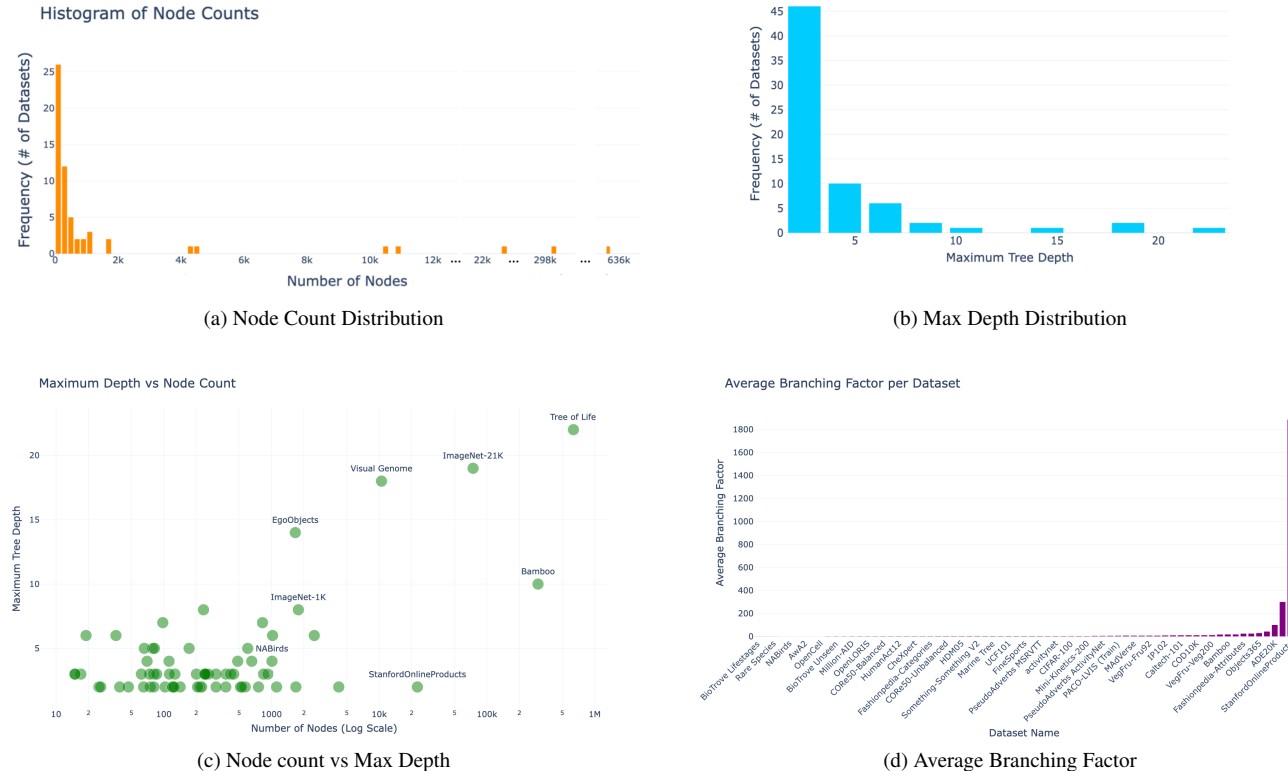

(a) Node Count Distribution

(b) Max Depth Distribution

(c) Node count vs Max Depth

(d) Average Branching Factor

Figure 2. Statistics and distributions of hierarchies in our *HierVision* collection.

ing. The loss is then the standard cross-entropy between the prediction and $\tilde{y}$:

$$\mathcal{L}_{\text{hier}-\text{smooth}} = -\sum_{j=1}^{N} \tilde{y}_j \log p_j \qquad (5)$$

where $p_j$ is the predicted probability for class $j$.

These hierarchy-aware objectives guide models to make semantically meaningful predictions, improve error robustness, and support finer-grained evaluation of learning performance.

### 4.2. Hyperbolic Learning

Hyperbolic learning uses the properties of hyperbolic space to better represent hierarchical relationships in vision datasets. Unlike Euclidean space, hyperbolic space expands exponentially, making it ideal for embedding tree-like taxonomies with minimal distortion. In practice, both class labels and image features are embedded as points in a hyperbolic manifold (e.g., the Poincaré ball), so that distances between points correspond to semantic or taxonomic proximity in the hierarchy.

For hyperbolic learning with vision datasets, a typical process has two main steps. Step 1 is embedding the hierar-

chy itself: the class tree is mapped into hyperbolic space so that classes which are close in the hierarchy are also close together in the embedding. This is usually done using methods such as Poincaré embeddings [88] or entailment-based approaches [36], which are specifically designed to preserve the distances and relationships from the original tree. The effectiveness of this embedding is measured by two metrics following the hyperbolic learning literature [88, 101, 103]: distortion, which captures how well the hyperbolic distances match the true tree structure (lower is better), and mean average precision (mAP), which reflects how well nearest neighbors in the embedding correspond to true neighbors in the hierarchy (higher is better).

Step 2, as done in works like [5, 57, 74], involves learning: projecting image features produced by a neural network into hyperbolic space and training them to align with their respective class embeddings. This allows the model to make predictions that are consistent with the structure of the hierarchy.

In this section, we focus on Step 1 and analyze how well different hyperbolic embedding methods can represent the hierarchy itself in Table 2. Using standardized trees from HierVision, we evaluate and compare several approaches

| | Poincaré [88] | | Entailment [36] | | BHE [57] | |
| --- | --- | --- | --- | --- | --- | --- |
| | Dist | mAP | Dist | mAP | Dist | mAP |
| CIFAR-100 [61] | 0.713 | 0.162 | 0.18 | 0.623 | 0.026 | 0.885 |
| ImageNet-100 [27] | 0.450 | 0.119 | 0.20 | 65.91 | 0.095 | 0.746 |
| CityScapes [24] | 0.540 | 0.173 | 0.250 | 72.41 | 0.050 | 0.967 |
| PASCAL-VOC [32] | 0.477 | 0.122 | 0.182 | 0.692 | 0.05 | 0.837 |

Table 2. Hyperbolic embeddings of CIFAR-100, ImageNet-100, Cityscapes and PASCAL-VOC in using different hyperbolic embedding methods. Distortion and mAP [101] of the embeddings measure how well the tree distances are embedded in the hyperbolic space.

on CIFAR-100, ImageNet-1k, ActivityNet, and CUB. For each dataset, we report distortion and mAP scores to assess how faithfully the class relationships are captured in hyperbolic space. We use the hyperparameters defined in Kasarla *et al.* [57] for generating the hyperbolic embeddings of the methods, keeping hyperbolic $dim = 64$. In Table 2, BHE [57] show better faithful tree embeddings. However, for downstream tasks, any of these embeddings can be used depending on the utility for the task.

The standardized JSON hierarchies in our hub make it easy to plug in such methods – for example, one can directly use NetworkX to compute ancestor relations or to feed the graph into a Poincaré embedding algorithm to obtain initial class vectors. The result is an integrated hyperbolic pipeline where both the data and the output of the model are easily integrated in the hyperbolic space.

**Note on end-to-end usage.** The *HierVision* hub provides reference training scripts for hyperbolic pipelines that are intended as reproducible starting points, instantiating common hyperbolic design choices on top of our standardized hierarchies (e.g., CIFAR-100, ImageNet-100, Cityscapes). Extensive empirical gains for hierarchy-aware and hyperbolic learning have already been reported elsewhere [3, 45, 57, 74, 116]; our contribution is to make the underlying hierarchical resources easy to find, load, and reuse.

## 5. Conclusion

We introduce *HierVision*, a hub for standardized hierarchical knowledge across a broad spectrum of visual recognition datasets. By consolidating and curating over 61 hierarchies from diverse domains and encoding them in a unified graph-based format, we provide consistent, reproducible access to structured label information. Our framework supports direct integration with graph libraries like NetworkX and enables hierarchical loss design, hyperbolic embeddings, and large-scale benchmarking. Recently, Ayoughi *et al.* [6] discussed optimal tree structure for hyperbolic embeddings, which can be further used to refine the existing hierarchies.

We believe that *HierVision* serves as a critical resource for the vision community, promoting reproducibility, accelerating hierarchy-informed research, and enabling rigorous benchmarking across a wide range of hierarchical structures. We shortlisted 40+ more hierarchies in the pipeline to be added in the future. We invite community contributions to further expand and refine *HierVision*, and we hope this hub will catalyze advances in hierarchical representation learning, benchmarking, and structured visual understanding at scale.

## Acknowledgements

We acknowledge partial support from the ELLIS Unit Amsterdam and the Data Science Center, University of Amsterdam. This work was also partially supported by the EU's Horizon Europe research and innovation programme within the ENEXA project (grant agreement no. 101070305).

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
