# OpenReview forum: "HierVision: Standardized and Reproducible Hierarchical Sources for Vision Datasets"
_thecvf.com/ICCV/2025/Workshop/BEW — BEW 2025 Oral_

### Official Review · Reviewer_FBi8 · 2025-07-02
**An extensive and well-presented source for hierarchical relationships in vision datasets**

**Rating:** 5
**Confidence:** 4

**Review:**

This work presents a unifying hub for hierarchical relationships in 60+ vision datasets across a wide range of imagery. In addition to collecting and unifying all of these sources, they also present a standardised format for storing the hierarchies that is compatible with standard graph processing frameworks. They provide visualisation of the hierarchies and an extensive overview of the datasets included in the hub.

I think this can be an important contribution to the non-Euclidean computer vision community. The paper does not make contribution in terms of methodology or applications, but still fits the scope of the workshop in terms of presenting modifications and improvements of taxonomies and datasets. As noted in the paper, hierarchical relationships are usually stored in widely different format, which makes them difficult to incorporate. Therefore, this type of standardisation is highly desirable, and can also act as a blueprint for applied researchers working outside publicly available benchmarking datasets. I also think they have done a lot of work to cover an extensive range of vision datasets. I also enjoy the visualisation of the corresponding hierarchies. In summary, I think this paper fits the workshop very well and can have a nice impact on the community, and I therefore recommend that it is accepted to the workshop.

---

### Official Review · Reviewer_1uyK · 2025-07-06
**Practical and useful contribution, lack end-to-end verification.**

**Rating:** 4
**Confidence:** 3

**Review:**

The main contribution of the work is a centralised repository of several (61) label hierarchies for vision tasks, formatted as a directed tree in JSON. The work posits itself as a resource for non-euclidean hierarchical representation learning, specifically targeting hyperbolic embeddings. The authors use NetworkX for verifying hierarchies, while providing some insightful exploratory data analysis to characterise the hierarchies. Figure 2 summarise some of the findings. Most hierarchies are quite shallow, with two- or three-levels, with a few datasets seem to push depths past ten. Node counts range from 20 to 500k leaf nodes / classes, and branching factors stay relatively low, with certain datasets (ADE20k, iNat, SOP) exploding. These are useful insights into the behaviour of common datasets under hierarchical modelling.

### HierVision for Non-Euclidean Modelling

The most relevant section for the workshop is section 4.2, where the authors look at four datasets; CIFAR-100, ImageNet-1k, ActivityNet, and CUB. Each dataset are embedded with Poincaré, entailment cones and BHE, with strongest results coming from BHE on CIFAR-100. However, these are limited to pure class embeddings; i.e. no images are projected, no hierarchy-aware classifier is trained, and no downstream gains are claimed. While this does align with the stated objectives of the paper, it comes across as a little underwhelming, since the proposed hierarchies are not demonstrated to be learned or estimated from the input data. A small demonstration of successful mapping of image to hierarchical labels for some of the smaller datasets would have significantly lifted the work, and demonstrated a semantic relation of the labels and input images.

HierVision does come with constraints that modellers need to factor into their workflow. Notably, all graphs are refactored as a rooted tree, and multiple-inheritance edges present in WordNet-style DAGs are removed, so any concept with two legitimate parents is reduced to one . That simplification streamlines code but could possibly discard relations that matter for semantic similarity. In practice, this may have certain consequences; e.g. if the label *"cello"* would be forced as a choice between *“string instrument”* and *“orchestral instrument”*, any distance-based metric could overstate the dissimilarity between *"cello"* and *"violin"* if the latter path is pruned. The manuscript could be improved to ensure modellers that such issues are not a concern, either by discussing cases or providing insight into how this pruning has been performed by the authors. As previously stated, an empirical verification would have gone a long way in demonstrating that the curated labels can be modelled with the input data.

The authors propose a roadmap with inclusion of approx. 40 additional hierarchies, and should be commended for their work in curating these datasets and adding hierarchical labels. In a supervised setting, this allows for hierarchy aware tree-based distances to inform the classifier, and Section 4.1.2 outlines hierarchy aware losses for applications with HierVision. In the more modern self-supervised settings, benefits are less clear; HierVision could possibly be used to verify hierarchies discovered with self-supervision, but the manuscript do not provide any discussion towards these applications.

### Summary

HierVision stands as a useful tool for supervised modelling in hierarchical vision tasks, and the authors have put in a lot of work in curating the data from multiple different sources and standardising the data. Some caveats such as normalisation to single inheritance is implicit as a result, and the manuscript is missing a end-to-end empirically verified example that clearly demonstrates that the labels can be applied in practical modelling from images. The authors provide structural verification through the process, but a semantic verification to put the dataset into practical use is currently somewhat missing.

While a repository of standardised hierarchical embedding of class labels is clearly a nice practical contribution, modellers will need to know if aligning individual images with the hierarchy is indeed feasible. This reviewer notes that the graphs are sourced from external sources, either dataset maintainers or other works, which somewhat removes the burden from the current authors. However, the concern is that that the standardisation choices such as single inheritance pruning could make modelling difficult for practitioners. The exploratory analysis performed by the authors somewhat helps to alleviate this concern, but additional guarantees would have made this a clear accept. This is the main concern of this reviewer towards the work. Still, this reviewer believes the contribution is solid enough to warrant an accept.

---

### Official Review · Reviewer_zdSz · 2025-07-07
**Review of HierVision**

**Rating:** 1
**Confidence:** 4

**Review:**

### Summary

The paper introduces *HierVision*, a curated collection of 61 hierarchical label files for vision datasets, converted into a common JSON graph schema and bundled with simple loading/visualisation utilities. While the repository will be handy, the manuscript itself offers little scientific novelty or evidence that the resource materially advances research.

---

### Strengths

* Freely aggregates hard-to-find hierarchies and standardises them in a single format.
* Clean, minimal code; one-line import via NetworkX lowers the barrier for adoption.
* Baseline hyperbolic-embedding scripts may serve as starting points for newcomers.

---

### Major Weaknesses

1. **Limited novelty** - The work is essentially data curation plus a thin wrapper library. Similar efforts (e.g. HuggingFace Datasets, torchvision loaders) are normally released as GitHub projects or blog posts. No new algorithm, theory, or substantive empirical finding is presented.
2. **Sparse empirical validation** - Only intrinsic embedding metrics (distortion, mAP) are reported. No downstream task (classification, detection, retrieval) is shown to benefit from the provided hierarchies.
3. **Maintenance & licensing unclear** - Long-term reproducibility hinges on versioning, governance, and dataset licences; these are acknowledged but not specified.
4. **Scalability** - Very large trees (≈600 k nodes) stored in flat JSON impose memory and I/O overhead. Binary alternatives (Parquet, HDF5) are not explored.

---

### Minor Comments

* The paper reads more like package documentation than a research article.
* Figures duplicate table content; space could be used for deeper analysis.
* Typos: “engingeering” → “engineering” (p. 4), “statstical” → “statistical” (p. 6).

---

### Suggestions for Improvement

1. Demonstrate end-to-end gains by plugging *HierVision* into a hierarchy-aware loss on ImageNet-100 or a fine-grained task.
2. Detail a maintenance plan (version tags, DOIs, community governance) and clarify licence compliance.
3. Compare storage formats and loading times for the largest hierarchies.
4. Consider submitting to a “datasets & benchmarks” track, where the novelty bar is different and a stronger quantitative analysis would suffice.

---

### Recommendation

In its current form, the manuscript does not meet the novelty threshold for a research paper. I recommend **rejection** and suggest releasing *HierVision* as a well-documented open-source project or resubmitting to a resource-focused track with more rigorous experiments.

---

### Decision · Program_Chairs · 2025-07-09

**Decision:**

Accept (Oral)

**Comment:**

The majority of the reviews agree towards accepting the paper.  The authors should do their best to address the comments of the reviewers in their final version.  The oral presentations would also present a poster.